A novel air quality monitoring and improvement system based on wireless sensor and actuator networks using LoRa communication

Truong Truong Van 1
Nayyar Anand anandnayyar@duytan.edu.vn 2
Masud Mehedi 3
1 Faculty of Electrical-Electronic Engineering and Institute of Research and Development, Duy Tan University , Da Nang , Viet Nam
2 Graduate School; Faculty of Information Technology, Duy Tan University , Da Nang , Viet Nam
3 Department of Computer Science, College of Computers and Information Technology, Taif University , Taif , Saudi Arabia
Tariq Muhammad
Electronic publication date: 2021 Sep 16
Publication date: 2021
Volume: 7
Electronic Location ID: e711
Received 2021 Jun 9; Accepted 2021 Aug 21
Copyright: ©2021 Truong et al.
Copyright year: 2021
Copyright holder: Truong et al.
License: This is an open access article distributed under the terms of the Creative Commons Attribution License, which permits unrestricted use, distribution, reproduction and adaptation in any medium and for any purpose provided that it is properly attributed. For attribution, the original author(s), title, publication source (PeerJ Computer Science) and either DOI or URL of the article must be cited.
License URL: https://creativecommons.org/licenses/by/4.0/

Keywords: Wireless sensor and actuator networks, Air monitoring and improvement system, LoRa/LoRaWAN communication, The Things Network, TagoIO, FreeRTOS

Funding: Taif University, Taif, Saudi Arabia, through Taif University Researchers Supporting under Project TURSP-2020/10 This work was supported by the Taif University, Taif, Saudi Arabia, through Taif University Researchers Supporting under Project TURSP-2020/10. The funders had no role in study design, data collection and analysis, decision to publish, or preparation of the manuscript.

==============================
In this paper, we study the air quality monitoring and improvement system based on wireless sensor and actuator network using LoRa communication. The proposed system is divided into two parts, indoor cluster and outdoor cluster, managed by a Dragino LoRa gateway. Each indoor sensor node can receive information about the temperature, humidity, air quality, dust concentration in the air and transmit them to the gateway. The outdoor sensor nodes have the same functionality, add the ability to use solar power, and are waterproof. The full-duplex relay LoRa modules which are embedded FreeRTOS are arranged to forward information from the nodes they manage to the gateway via uplink LoRa. The gateway collects and processes all of the system information and makes decisions to control the actuator to improve the air quality through the downlink LoRa. We build data management and analysis online software based on The Things Network and TagoIO platform. The system can operate with a coverage of 8.5 km, where optimal distances are established between sensor nodes and relay nodes and between relay nodes and gateways at 4.5 km and 4 km, respectively. Experimental results observed that the packet loss rate in real-time is less than 0.1% prove the effectiveness of the proposed system.

Introduction

In recent years, the problem of air pollution in residential areas and factories is becoming more severe under the impact of six factors: Nitrogen Oxide (NOx), Sulfur Oxide (SOx), Carbon Monoxide (CO), lead, ground ozone, and dust. Among them, fine particles (PM2.5) with a size of fewer than 2.5 microns cause the most serious consequences, as they can penetrate deeply into the lungs, affecting both the respiratory system and circulatory system (Wang et al., 2020). According to the World Health Organization (WHO) statistics, annually, more than 90% of people are exposed to outdoor concentrations of PM2.5 that are higher than the air quality standards. According to the study (Liu et al., 2020), whether people’s health status is serious or not will depend on the degree and time of exposure to the polluted air. In addition to its negative impacts on the environment and human health, air pollution also reduces productivity and reduces energy efficiency. Several studies have demonstrated an increase in CO and CO2 levels, leading to an increase in the amount of volatile organic compounds (VOCs), odors, and microorganisms in the air (Sadatshojaie & Rahimpour, 2020). That makes a decrease in humans’ ability to concentrate. Furthermore, according to Franco & Schito (2020) and Franco & Leccese (2020) study, controlling CO and CO2 concentration in the air can lead to up to 5% to 20% energy savings in HVAC systems in buildings.

Several air quality monitoring systems based on wireless sensor network (WSN) (Dhingra et al., 2019; Marques, Ferreira & Pitarma, 2019; Yi et al., 2015; Mansour et al., 2014; Arroyo et al., 2019) have been proposed with such urgent problems. These systems include sensors scattered throughout the sensing area to collect air quality parameters such as dust concentration or exhaust gas concentration. Data is collected at the central station over a wireless transmission link, processed by the software and analyzed for air quality. Also, they can provide relevant recommendations or build a visual air quality map. However, the most significant limitation of these systems is that real-time air quality improvement in the monitoring area has not been resolved because they are not equipped with appropriate control actuators.

Therefore, a wireless sensor and actuator network (WSAN) has been proposed as the next generation of development of WSN (Verdone et al., 2010; Pitarma, Marques & Ferreira, 2017; Bai et al., 2017; Primeau et al., 2018). WSAN is making significant progress thanks to IC technology, sensor technology, and wireless communication technology. The components of WSAN are the sensor node, sink/actuator node, and gateway. Sink/actuator nodes are often more involved in structure than sensor nodes because besides monitoring environmental parameters, they are also equipped with actuators to react to their surroundings under precise conditions (Verdone et al., 2010). The sensor node has a more straightforward structure and more compact size and is usually powered by a convenient battery for deployment in real-world environments. The gateway is a device that collects all the parameters of the system and sends it to the Internet server for signal processing and storage purposes.

Besides the challenges of hardware deployment of WSAN systems, software and perfecting communication protocols in sensor networks are also very complex (Pitarma, Marques & Ferreira, 2017; Nayak & Stojmenovic, 2010). A series of wireless communication techniques have been proposed for WSAN for indoor and outdoor surveillance applications, listed as Bluetooth, Zigbee, LoRa, NB-IoT, Z-wave (Saini, Dutta & Marques, 2020; Arroyo et al., 2020; Chaturvedi & Shrivastava, 2020; Liu & Bi, 2017; Abraham & Li, 2014). For short range environmental monitoring applications, Bluetooth and Zigbee are two suitable techniques. Sensor nodes equipped with Bluetooth communication operate at 2.4 GHz over a maximum distance of 75 m, with peak data rate with EDR is 3 Mbps (Saini, Dutta & Marques, 2020; Arroyo et al., 2020). Furthermore, Bluetooth also supports three modes of security with non-line-of-sight communication. Zigbee is also a short-range communication technique at 2.4 GHz. Zigbee is suitable for WSAN applications that do not require high transmission rates, low cost, and low energy consumption (Chaturvedi & Shrivastava, 2020; Liu & Bi, 2017; Abraham & Li, 2014). In WSAN’s vast area network, LoRa technology is considered the most optimal solution. LoRa operates at the frequencies 433, 868, and 915 MHz and can achieve very low-power and very long-range transmissions (over 10 km in line-of-sight) (Alvear-Puertas et al., 2020; Ma et al., 2018; Botero-Valencia et al., 2018; Truong, Nayyar & Showkat, 2021). LoRa is suitable for low bandwidth and relatively small payload applications.

Moreover, the WSAN model for air quality monitoring and improvement system (AQMIS) will provide the same benefits and gain more advantages. Typically, in an enclosed space indoors, people tend to use exhaust fans to increase the airflow in and out of the house to improve air quality. However, the air quality index, negative ion concentration (Liu et al., 2021; Marć et al., 2015; Gandhi et al., 2020) cannot be improved by this method. Negative ions have been widely used worldwide thanks to their superior properties such as sufficient anti-allergy ability, smoke-free, 52% dust reduction, 95% reduction in bacteria in and reduce airborne pollutants by up to 97% within a distance of 1.5 m (Marć et al., 2015). The reason is that impurities in the air such as smoke, dust, etc. are usually positively charged; negative ions will attract these positively charged particles, neutralize them, and fall to the ground, helping the air become fresh and clean. For humans, the cell contains many mineral ions involved in bio-electric processes. Negative ions in the air will participate in this process to help increase metabolism, increase oxygen in the blood and support the collective nervous system to function better. However, the problem is that negative ions only exist in mountainous areas, coasts, suburbs, green parks (10.000 to 500.000 negative ions/cm3), while indoors, their content is only about (0 up to 40 negative ions/cm3). Therefore, the design of a negative ion generator is essential to the realization of WSAN-AQMIS.

To the best of our knowledge, there are currently no works on the application of WSAN based on the two layers LoRa/LoRaWAN in AQMIS, so it motivates us to do this study. Our proposed model well solves the disadvantages of LoRaWAN: it does not support node-to-node communication. Moreover, communication layering also enhances the scalability of the system. The main contribution of our work is as follow:

• The two layers WSAN based on LoRa/LoRaWAN communication for AQMIS are proposed, which contains two sub-systems: indoor and outdoor. The system can connect to many different sensors such as dust, CO, LPG, and CH4 concentration sensors. The system is expandable due to the operation of the full-duplex relay LoRa module. We proposed using LoRa wireless communication technology for node-to-relay layer and LoRaWAN, a network protocol using LoRa, for relay-to-gateway layer.

• The designing negative ion generators as actuators are proposed at the nodes with igh efficiency and operating continuously for a long time.

• The ADC noise reduction mode and digital Kalman filters are proposed to enhance the reliability of the air parameter measurements. In addition, we proposed using the real-time operating system FreeRTOS to manage tasks for the full-duplex relay LoRa module.

• The design the software monitoring parameters on the web and smartphone interface using The Things Network (TTN) and TagoIO platform are proposed.

This paper is organized as follows. In ‘Literature Review’, we introduce the literature review. In ‘Method and Procedures’, the hardware and software of the proposed system are described. In ‘Experimentation and Results’, we evaluate the system performance with the critical parameter. And the last one, we present the conclusions in ‘Conclusion and future work’.

Literature Review

In this section, we present studies related to the wireless sensor and actuator network and air quality monitoring and improvement system.

Dhingra et al. (2019) proposed a WSN based on WiFi communication with the hardware of sensor nodes, cameras, and Android data monitoring applications. Sensor nodes detect harmful gases such as CO2, CO, …and the camera will collect traffic data in the city. Data stored at Cloud servers and Android applications will help users identify routes with high levels of air pollution and choose suitable routes to move. However, the authors have not mentioned dust - the leading pollutant at present in their work. Although the Android application helps users find less polluted routes to transport, the interface is simple, not meeting the need for visual monitoring through graphs. The study also did not specify that the most critical parameters of Wifi-based WSN, i.e., the distance of data transmission, network coverage, the lifetime of the sensor node. Furthermore, a WSN model that is entirely dependent on Wi-Fi is not reliable in practice.

Marques, Ferreira & Pitarma (2019) proposed a model for indoor air quality monitoring. The system uses an MHZ-16 sensor to measure CO2 indoors and uses the ESP8266 module to transmit data to the web-server. The IAirCO2 application helps users to recognize the pollution level where the sensor is located. However, the defect of the application interface is still relatively simple, not meeting the graph’s visual monitoring need. System customization and scalability are also limited because the design is based on Sparfkun’s existing hardware.

Arroyo et al. (2019) proposed a low-cost, low-size, low-power consumption real-time monitoring air quality system. The sensor nodes communicate with the master station via Zigbee communication. An optimized fog computing system has been performed to store, counselor, process, and imagine the sensor network’s data. Data processing and analysis are implemented in the Cloud by applying artificial intelligence techniques to optimize compounds and contaminants. Finally, the authors use a simple case study to prove the algorithm’s effectiveness in detecting and classify harmful emissions.

The Zigbee network is proposed by Abraham & Li (2014) for indoor air quality monitoring. The hardware model is deployed indoor with four sensor nodes measuring air quality using the Zigbee network deployed by Arduino and XBee transmission module. Sensor nodes detect harmful gases parameters and sent to the base station for storing. The author is elaborate on processing measured data, and the parameters are represented in the form of objective graphs. However, the system’s applicability is limited because the system only monitors indoor air quality and does not mention other polluting agents such as dust, bacteria in the house. Moreover, the author is not clear about the data transmission distance and protocol of Zigbee communication.

LoRa communication has been proposed and applied in many designs to solve the limited transmission distance problem of Wi-Fi, Bluetooth, or Zigbee. Alvear-Puertas et al. (2020) proposed a model to monitor parameters CO, NO2, PM10, PM2.5 using an STM32F107 microcontroller. The system consists of a sensor node that communicates with one server via LoRa communication. The authors have conducted many experiments measuring environmental parameters and comparing them with the national control station’s standard data. As a result, the system can operate well within 2 km and an error of 5 to 8%. However, the proposed system model is quite simple and does not meet a wide-area WSN system’s requirements.

Firdaus, Murti & Alinursafa (2019); Wang et al. (2017) present a similar model of using LoRa networks in air quality monitoring. In addition to the parameters of the level of environmental air pollution, the authors also examine the issues of battery life and time delay during communication. However, the authors have not given the survey when the system works with many sensor nodes and communication protocols to ensure the reliability of the data.

We summarize some studies on air quality monitoring systems and related specifications in Table 1.

Table 1 The AQMIS and relevant parameters in existing studies.

Reference	Parameter	Connectivity	Type	Packet loss	No. of nodes	
Abraham & Li (2014)	CO, CO2, volatile organic compounds	Zigbee	Indoor	NM	6	
Arroyo et al. (2019)	Gas concentration	Zigbee	Outdoor	NM	NM	
Botero-Valencia et al. (2018)	Gas concentration, PM2.5	LoRa	Outdoor	NM	1	
Chaturvedi & Shrivastava (2020)	CO2, SO2, NO2	Zigbee	Indoor	NM	1	
Dhingra et al. (2019),
Marques & Pitarma (2019)	CO, CO2, CH4	Wifi and GPS	Outdoor	NM	NM	
Firdaus, Murti & Alinursafa (2019),
Zhao, Wu & Li (2019)	CO, CO2, Temperature, Humidity	LoRa	Outdoor	6%	1	
Mansour et al. (2014)	CO, CO2, NO2, CH4, NH3	Zigbee	Outdoor (500 m)	NM	8	
Pitarma, Marques & Ferreira (2017)	Gas concentration	Zigbee	Indoor (50 m)	NM	2	
Marques et al. (2019)	Gas concentration	Bluetooth	Indoor (50 m)	NM	1	
Notes.

NM Not mentioned

Most of the studies above have focused on air quality monitoring using WSNs with different transmission techniques. However, their most significant limitation is that the proposed model is quite simple and does not meet the practical requirements of WSN (short distance communication, few number of sensor node). The main system-specific parameters such as communication distance, packet loss rate have also not been investigated. Furthermore, the issue of improving air quality has not yet been investigated. Therefore, we propose a WSAN model for air quality monitoring and improvement using LoRa/LoRaWAN communication in this study.

Method and Procedures

System model

In this part, we present the proposed model of the WSAN system for indoor and outdoor air quality monitoring and improvement (WSAN-AQMIS) application using LoRa communication. The system consists of three parts: Indoor sensor cluster, Outdoor sensor cluster, and gateway which communicate through LoRa communication. The sensor nodes send information to the gateway using uplink LoRa, while the gateway controls the actuators through the downlink LoRa. Also, we use TTN to build real-time online data management software for our proposed system.

Figure 1 shows the proposed system model. Specifically, the indoor cluster includes four sensor nodes that monitor air quality parameters, including dust concentration, CO concentration, and GAS concentration. The indoor sensor node can connect to Wi-Fi with the Wi-Fi router and instantly transmit local data to the Blynk server. These nodes also feature integrated functions that can communicate with the gateway using the 433 MHz uplinks LoRa. We also designed the outdoor cluster to provide long-range air quality monitoring for the system. The two outdoor sensor clusters are stratified into the outdoor node and the full-duplex relay node. Each outdoor sensor node will have an integrated solar collector responsible for collecting temperature, humidity, air quality parameters and transmitting information to the relay node. The relay nodes have a built-in function that forwards sensing data to the gateway. Depending on air quality, the gateway will make decisions to control the actuator using downlink LoRa. The actuators in our system are divided into two categories for indoor or outdoor equipment. In the indoor node area, we are equipped with an exhaust fan, and a negative ion generator (NIG) to enhance the air quality. On the outdoor, nebulizer controls will be fitted. These devices help to increase air humidity and clean the dust in large spaces. We do not use NIG outdoors due to the disadvantages of the power supply and the limited capacity of the device.

Figure 1 The system model of the wireless sensor and actuator network for air monitoring and improvement application.

Architecture

In this section, we present the hardware design and technical working of the proposed system.

The indoor cluster consists of four sensor nodes with two modes of operation. In the first mode, when the Wifi or LoRa transmission is inoperative, the nodes will communicate through ESP-NOW. It is a protocol developed by Espressif that helps nodes connect peer-to-peer does not require handshakes. In the second mode, the nodes can communicate directly with the Blynk server via Wifi and connect to the gateway via LoRa communication. Although long-range is not required for indoor nodes, using the Lora-based for the whole system helps the data streams received by the gateway to have the same format and easily be processed. Furthermore, implementing LoRa makes it easy for us to perform Over-The-Air-Activation (OTAA), which is downloading new firmware to the ESP32 via wifi instead of using a traditional Serial port.

The Indoor nodes; as Fig. 2; are distributed scattered in a narrow range such as households, offices, classrooms. We use the central processor, LoRa32 module manufactured by Heltec, which has a built-in LoRa communication module and peer-to-peer communication from the ESP32. It receives energy directly from the grid and communicates with dust, gas concentration, temperature, and humidity sensors. Furthermore, the module can work well with many different types of sensors in the MQ brand, i.e., MQ2 to MQ9 family. The LoRa32 module is designed with a 0.96-inch OLED screen capable of displaying all collected sensor parameters. Besides, the LED indicator system is designed to show four pollution levels in the monitoring area according to standards: Good (Air quality index - AQI < 50); average (51 < AQI < 200)l poor (201 < AQI < 300); hazardous (AQI > 300). Depending on the control signal received from the gateway, the indoor node will control two corresponding actuators, the exhaust fan and the negative ion generator.

Figure 2 Indoor node schematic.

The negative ion generator circuit principle comes from using a very high voltage source to ionize the air as Fig. 3. We proposed the high voltage generator contains a NE555 timer IC to generate square-wave pulses. The pulses are implemented to the base of the TIP120 Darlington transistor. This design ensures the power transistor 2N3055 is supplied with enough current to turn it on. When the transistor is open, current flows through the high voltage auto-transformer T2 and is connected to a 10 kV high voltage diode IMD5210. The diode is biased to negatively charge C3 and C4, giving the discharge point charged. Air is continuously blown to this point thanks to the synchronous dynamic movement of the exhaust fan.

Figure 3 Negative ion generator schematic.

The outdoor cluster consists of four sensor nodes divided into two subgroups supported by two relay modules. The outdoor sensor node is designed to operate in long-term outdoor conditions; in addition to the function of monitoring the environmental parameters, they are also equipped with a solar collector module and water resistance as seen in Fig. 4. The device is equipped with a single-cell Lithium battery solar charge controller IC CN3065 capable of automatically powering solar panels and managing charging current for a Lithium battery. The 5V and 3.3V linear source ICs provide a steady DC voltage to the circuit.The heart of the device is the Atmega328P from Microchip and the LoRa Ra-02 module that communicates with each other by SPI. We use class A for sensor nodes to save energy. Class A is usually applied to nodes that use batteries and send data through the gateway at any time. Sensors allow the system to monitor a wide range of environmental parameters such as temperature, humidity, dust, CO and LPG concentrations. These parameters are visually displayed on the I2C OLED screen and transmitted to the LoRa transmitter. The actuator controlling signal is received via the downlink LoRa based on the collected sensor information.

Figure 4 Outdoor node schematic.

A critical component of the system is the full-duplex LoRa relay module as Fig. 5, capable of extending the system’s coverage. According to TTN Fair Access Policy, the maximum number of nodes that this gateway can support is: (1) n=nf.ns.dcT=1.86400.1%30=28

where nf is the number of frequency bands that the gateway supports, ns is the number of seconds in a day, dc is duty cycle, T is the air time per device per day.

Figure 5 Relay node schematic.

While the communication range between LoRa modules is excellent, we still recommend using this module for several reasons. First, two LoRa modules work together in a transceiver to make the system easy to expand while not causing additional communication delays. One LoRa module will act as the receiver, while the other will act as the transmitter; they are managed by SPI interface with microcontroller and operate by interrupt mechanism. The second reason, if the system operates in half-duplex mode, two LoRa modules simultaneously receive packets and ensure that the system’s packet loss rate is reduced. After receiving all the packets in the cluster they manage, an idle LoRa module converts the task into a transmitting module to send signals back to the gateway. This implementation method ensures a low packet loss rate but increases the communication delay time.

In this design, we use the open-source Dragino LG01N device energy-saving module as the LoRa gateway as Fig. 6. Dragino forwards LoRa signals from sensor nodes to Cloud Server via WiFi, Ethernet, 3G, or 4G links. With a CPU HE Linux module 400 MHz, 64MB RAM, 16MB Flash connected to LoRa module SX1278 via SPI interface and packaged according to industry standards, LG01N has a super far data transmission range (5–10 km) and outperform anti-interference. Besides that, the built-in 2.4 GHz WiFi card, EC25 LTE SIM card, and two network port RJ45 LAN - RJ45 WAN in LG01N provide many methods of connecting to high-speed Internet. LG01N supports LoRaWAN protocol at various frequencies and can customize different encoding techniques such as FSK, GFSK, MSK, GMSK, LoRaTM, and OOK. The LG01N can achieve very high sensitivity, at −148 dBm, and serves very well from 50 to 100 sensor nodes. In addition to the LoRa to Cloud Sever wireless signal conversion, LG01N can support many working modes such as MQTT mode, TCP/IP Client Mode, to suit different IoT connections requirements. Another advantage is that LG01N can be firmware upgraded directly from the Web GUI environment without any dedicated loader’s assistance.

Figure 6 The LoRa gateway Dragino.

Software

This section highlights the algorithm of the proposed system.

Figure 7 describes the indoor node algorithm flowchart. The MCU sets the necessary input/output parameters for the system and also specifies the ESP-NOW protocol. We used LoRaWAN as the default protocol for the proposed system, but in the case, LoRa modules do not guarantee communication, the ESP-32 is set up to communicate peer to peer using the MAC address thus improving system reliability. In case the user wants to use Blynk for on-the-spot surveillance with a smartphone without going through the gateway, the Blynk server will be set up. In case the ESP-NOW protocol is enabled, indoor sensor nodes can communicate with each other directly via the super energy-saving 2.4 GHz channel. ESP-NOW has built-in sending and receiving callback functions to secure communication, so it is a reliable protocol. Before transmitting, the nodes will perform a MAC address pairing operation. After pairing is complete, the devices have established a network and can communicate with peer-to-peer. If a node in the network connects to the Dragino gateway, the network data will be automatically sent. Data is collected at the nodes using the “Read sensor data” module. Sensor information is collected by the ADC module integrated into the micro-controller. We use the Kalman Filter, a powerful digital filter that combines current uncertainty with environmental noise into a new, more reliable form of information for future prediction. The strength of Kalman filter is very fast running and high stability. Furthermore, we deploy ADC noise reduction, an internal noise reduction mechanism in the microcontroller by limiting the operation of the IO module, to increase measurement accuracy. With the help of these two techniques, the nodes ensure accurate data measurement.

Figure 7 The algorithm flowchart of the indoor node.

The gateway’s feedback is an air quality index (AQI) value that helps sensor nodes perform on/off actuators. The procedure for calculating the AQI of the parameters SO2, CO, NO2, PM10, PM2.5 is as follow: (2) AQIx=Ii+1−IiBPi+1−BPiCx−BPi+Ii

where:

• AQIx is the air quality index of x parameter.

• BPi is the lower limit concentration of observed parameter value specified in each region/country corresponding to the level i.

• BPi+1 is the upper limit concentration of observed parameter value specified in each region/country corresponding to level i + 1.

• Ii is the AQI value at level i following the BPi value.

• Ii + 1 is the AQI value at level i + 1 following the BPi+1 value.

• Cx is specified as follows: For PM2.5 and PM10 parameters, Cx is the average value collected after 24 h. For SO2, NO2 and CO parameters, Cx is the average maximum value of one hour per day.

After having AQIx value of each parameter, the maximum value is chosen to be aggregated AQI value according to the following formula: (3) AQI= maxAQIx

Notice the aggregated AQI value is rounded to an integer. The outdoor node’s operation is similar to the indoor node but does not include the ESP-NOW protocol and focuses on energy saving. Figure 8 depicts the outdoor node operation algorithm flowchart. First, the system sets the Input/Output ports, SPI ports for the LoRa module, the Interrupt services with low priority, and the Deep sleep mechanism with high priority. Deep sleep and periodic wake-up mechanisms for the outdoor node ensure that the outdoor nodes consume the lowest power levels. We recommend using an interrupt service routine for receiving control data from the LoRa relay. After receiving control data, the MCU controls the respective actuators’ opening/closing to improve the air quality in the surveillance area.

Figure 8 The algorithm flowchart of the outdoor node.

Figure 9 shows the algorithm flowchart for the full-duplex LoRa relay module. We proposed embedding the FreeRTOS real-time operating system with MCU to ensure simultaneous sending and receiving tasks (Kampmann et al., 2019; Docekal & Slanina, 2017). FreeRTOS is an open-source Real-Time Operating System developed by Real Time Engineers Ltd. FreeRTOS is designed with straightforward functionality such as basic task and memory management, synchronization API functions, with a total size of only 4.3 KB. Here, the MCU performs three tasks simultaneously with decreasing priority as follows. The first Task, which has the highest priority, ensures that LoRa data is always listened to by the LoRa modules. The second task, which has a lower priority, takes on the role of the LoRa transmitter. Thus, two LoRa Ra02 modules controlled by RTOS ensure the information acquisition and forwarding of received information to the gateway smoothly. The full-duplex protocol allows the system to operate without additional communication delay. However, the trade-off is that the packet loss rate is difficult to guarantee because only one module does the LoRa data collection service. Moreover, the last task, which is the lowest priority, is optional and ensures the visual mechanisms on OLED for debugging. We use Prioritized Pre-emptive Scheduling with Time Slicing algorithm to determine which tasks should be put into the Running state (Trivedi, 2014). Thus, the task in the Ready state with the highest priority will be executed first or take over the execution of the running task. These data use the same data resources and are controlled and synchronized using the Mutex binary semaphore. A mutex can be viewed as a token and assigned to a data resource. Whenever a task wants to access a resource, a token must be held. Then other tasks will be queued until the token is released.

Figure 9 The algorithm flowchart of full-duplex relay node.

Management software

In this section, we discuss about the structure of the management software of WSAN-AQMIS on the The Things Network (TTN) and TagoIO tool.

The Things Network is a global collaborative Internet of Things network that allows all members to bring their network together like one big Internet. The main difference between TTN and earlier networks is that it is a community project that does not depend on any corporate networks. That means developers can build their applications completely independently and get great support from the open LoRa community, independent of other service providers. Furthermore, users contribute ports to upload data from LoRaWAN sensor nodes to the Internet. According to their open standards, TTN grow extremely fast, with an estimated 10,000 LoRaWAN gateways in 147 countries currently connected through gateway operators using management and security tools.

TagoIO is a tool that can be directly connected to TTN thanks to authorization by device-token and used to design a dashboard. TagoIO dashboard contains widgets that help users efficiently observe and manipulate real-time data. All data are stored in Data Buckets. Once the LoRa device is connected to TTN, TagoIO will create a bucket to hold the corresponding data. This study designed a dashboard consisting of two tabs to manage the indoor sensor cluster and outdoor sensor cluster. The Indoor sensor nodes tab displays parameters of dust concentration, gas concentration, temperature, and humidity. These parameters are represented as graphs as follows:

• Gauge chart: display instant information of environmental parameters

• Vertical and horizontal bar graphs: information 5 and 10 nearest signal samples are displayed; the higher the value, the higher AQI.

• Time chart: graph of parameters on the 1-hour scale.

The user can use the node option to access the Dashboard of the node of interest. The charting functionality is similarly designed for the Outdoor sensor nodes tab. Figure 10 depicts the live data of indoor node ID 1, while Fig. 11 depicts the data at two outdoor nodes managed by a relay. A mighty function that we exploit in TagoIO is data statistics. The scripts that run at TagoIO are programmed with Node.js. We use scripts to calculate AQI, export reports, and control the downlink actuator with the Actions tool. Data is stored and exported to any email as CSV format. All designs on the dashboard are synchronized on the real-time smartphone interface as Fig. 12. Besides, the indoor monitoring system can work well thanks to the ESP-NOW protocol and monitor by Blynk when there is no WIFi network. Two dashboards are active at the same time ensure that the data monitoring process is always reliable. Furthermore, the Blynk dashboard is a reference tool to debug the system, thanks to its friendly user interface support.

Figure 10 The Tago dashboard intergrated with TTN for the indoor sensor node.

Figure 11 The Tago dashboard intergrated with TTN for the outdoor sensor node.

Figure 12 The Tago dashboard on smartphone and Blynk dashboard.

Experimentation and Results

In this section, we evaluate system performance in various scenarios using the parameter of packet loss rate. We set the experimental parameters in Table 2.

Table 2 Simulation parameters.

Parameters	Notation	Typical values	
Number of users in indoor cluster	M	4	
Number of users in outdoor cluster	N	2	
Number of relay node	K	2	
Bandwidth	BW	125 kHz	
Code rate	CR	1	
Spreading factor	SF	7	
Number of payload frame	P	500	
Payload length	L	30 byte	
Distance from the relay node to the gateway	d rg	3.5 km	
Distance from indoor nodes to the gateway	d ig	20 m	
Distance from outdoor nodes to the relay	d nr	4 km	

The completed hardware deployed in work is shown in Fig. 13, positioned with nodes as Fig. 14. The steps to perform the experiment are as follows:

Figure 13 The complete model of the proposed system.

Figure 14 The location of the nodes is displayed in TagoIO.

• Determine experimental model: In this step, we select the devices participating in the experiment.

• Determine evaluation parameters: we only choose 1 or 2 parameters to be investigated for each experiment; other parameters are kept fixed.

• Conduct experiments: carry out the survey 50 times for each experiment and take the average value for the measurements.

In the first test scenario, we examine the packet loss rate according to communication distance. The scenario experiment consists of two outdoor sensor nodes, one full-duplex relay node, and one gateway. Gateway is permanently located close to the Wifi router so that the dashboard can operate stably. Figure 15 depicts the packet loss rate with increasing distance from the outdoor node to the relay node. We set the distance between the relay node and the gateway drg fixed to be 3.5 km. In each experiment, the outdoor sensor nodes send 500 frames of numbered data to the gateway at bandwidth BW is 125 kHz, code rate CR is 1. Each payload frame long 30 bytes. The dashboard will receive and record the number of packets lost, duplicated, and incorrectly formatted during this communication. We increase the distance between the outdoor node and the relay dnr from 100 m to 6 km and measure the packet loss rate. The obtained results show that the system works well in the distance of 4 km and the rate packet loss is less than 1%. We continue to investigate the impact of FreeRTOS in the recommendation system. Using FreeRTOS ensures the full-duplex relay node transmits and receives data simultaneously and minimizes the possibility of data conflicts on the transmission line, thereby improving system performance. The experimental results show that using FreeRTOS at the relay node can reduce the packet loss rate to less than 0.01% over a distance of 4.5 km. Moreover, using FreeRTOS significantly improves the communication distance. It can be explained as follows: When not using FreeRTOS, Ra01 modules receive transmissions from nodes when the LoRa signal has a good enough signal-to-noise (SNR) ratio. In case the communication distance is too large, the two Ra01 modules on the relay node get the internal communication interference with each other because the distance between them is very close. Meanwhile, using FreeRTOS helps the system divide tasks according to available hardware, thereby eliminating internal communication interference, increasing SNR, and helping the system operate over longer distances. After 50 times of this experiment, we recommend the optimal location for the outdoor sensor nodes with relay node to be 4 km in the case not use RTOS system and 4.5 km when use the RTOS system.

Figure 15 The packet loss rate vs. the distance from the outdoor node to relay node.

Figure 16 depicts a similar experimental scenario, with the distance between two outdoor nodes and the relay (dnr) is fixed at 4 km and gradually increases the distance between the relay node to the gateway. All system parameters are kept as in the first scenario. The results once again confirm the use of FreeRTOS can reduce the number of packet loss in the proposed system. We also recommend that the distance between the relay node and the gateway is 3.3 km in not using the RTOS and 4 km in using the RTOS.

Figure 16 The packet loss rate vs. the distance from relay node to gateway.

In the next experiment, we investigate the effect of the payload on the transmission speed, expressed by the time on-air (ToA) parameter as Fig. 17. The experiment scenario consists of an outdoor sensor node, a full-duplex relay node, and a gateway. The system parameters are set as follows: BW = 125 kHz, CR = 1, dnr = 4 km, drg = 4 km. In turn, we set up different spreading factor (SF) values (from 7 to 12) and increment the payload in this experiment. The results shown in Fig. 17 demonstrate that the higher the payload, the higher the ToA, which means the slower the transmission speed. Another observation is that the higher the SF, the higher the ToA. It concludes that a high SF can cause great latency; however, a high SF will be used in some cases where an increase in coverage is desired.

Figure 17 The time on air vs. different payload.

We continue to investigate indoor air quality improvement for the entire system with the parameters given in Table 2. The parameters are collected in 24 consecutive hours and statistically by Dashboard TTN in 2 cases: (i) system does not use the actuators and (ii) system uses such devices. The results show that using a negative ion generator and exhaust fan significantly improves air quality, especially for areas with narrow space as Table 3.

Table 3 Indoor AQI improvement vs time.

Time	Location	Case (i)	Case (ii)	Time	Location	Case (i)	Case (ii)	
00:00 to 05:59	Kitchen	8	4	12:00 to 17:59	Living room	10	4	
06:00 to 11:59	Kitchen	21	4	18:00 to 23:59	Living room	8	2	
12:00 to 17:59	Kitchen	16	4	00:00 to 05:59	Bedroom	6	1	
18:00 to 23:59	Kitchen	13	4	06:00 to 11:59	Bedroom	8	2	
00:00 to 05:59	Living room	6	1	12:00 to 17:59	Bedroom	7	2	
06:00 to 11:59	Living room	14	4	18:00 to 23:59	Bedroom	8	2	

Conclusion and future work

This work presents the AQMIS based on WSAN. The proposed system involves four indoor nodes that operate synchronously with two different protocols, i.e., LoRaWAN and ESP-NOW, under a gateway’s management. Also, the outdoor sub-system cluster allows for remote air quality monitoring. The system has extensive coverage thanks to the full-relay LoRa module’s operation under the real-time operating system FreeRTOS. Furthermore, we apply ADC noise reduction and Kalman filter to increase measurement accuracy. The system is monitored online via a dashboard based on TTN and TagoIO. The control signals are automatically fed back to the corresponding actuator through the downlink LoRa. The experiment results show that the system performance is highly achieved and capable of practical applications. The amount of air quality data that we have collected over the past three months is of great value in environmental management.

In future work, we will continue to expand the system by equipping mobile for indoor nodes. Besides, we will install more outdoor nodes for the planning of air quality maps.

Supplemental Information

Supplemental Information 1 Code–Detailed Code for Sensor Nodes

Click here for additional data file.

Supplemental Information 2 Layout file for Indoor node

Click here for additional data file.

Supplemental Information 3 Layout file for Relay node

Click here for additional data file.

Supplemental Information 4 PCB picture of Indoor node

Click here for additional data file.

Supplemental Information 5 Layout file for Indoor node

Click here for additional data file.

Supplemental Information 6 PCB picture of Outdoor node

Click here for additional data file.

Supplemental Information 7 PCB picture of Relay node

Click here for additional data file.

Additional Information and Declarations

Competing Interests

Author Contributions

Patent Disclosures

Data Availability

Anand Nayyar is an Academic Editor for PeerJ.

Truong Van Truong conceived and designed the experiments, performed the experiments, performed the computation work, prepared figures and/or tables, authored or reviewed drafts of the paper, and approved the final draft.

Anand Nayyar conceived and designed the experiments, performed the experiments, analyzed the data, performed the computation work, prepared figures and/or tables, authored or reviewed drafts of the paper, and approved the final draft.

Mehedi Masud conceived and designed the experiments, analyzed the data, authored or reviewed drafts of the paper, and approved the final draft.

The following patent dependencies were disclosed by the authors:

Name of patent: LoRa - Zigbee Hybrid Smart Communication System

Patent No: 2020102711, Australian Patent

Effective date of patent: 2020-10-14

Name of the author: Anand Nayyar, Van-Truong Truong

The following information was supplied regarding data availability:

The codes are available as Supplemental Files.

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
