# Peer review of "A novel air quality monitoring and improvement system based on wireless sensor and actuator networks using LoRa communication"

_PeerJ Computer Science, doi:10.7717/peerj-cs.711_

## Round 0.1 · original submission · Major Revisions

I find it difficult to recognize the contribution to the body of knowledge in this work. The authors need to justify the novelty of the manuscript.
Why are the authors used a schematic diagram for the indoor node in Figure 2?

Is PCB design by authors themselves? If not then a proper reference should be provided?

There are various state-of-the-art air quality monitoring systems are developed. The authors should draw a comparison with the existing state-of-the-art works.

LoRa is used for long-range (up to 2 KM range) and low data rate communication. The authors should justify why they have used it in indoor communication where longer range is not always desired.

Reviewer 1 ·

Basic reporting

n/a

Experimental design

n/a

Validity of the findings

n/a

Additional comments

The Research Paper needs the following Minor Revisions and is subject for the following revisions and after revisions, the paper is recommended for re-review:
1 Make sure you use the same style of notation and make sure all be consistent.
2 Thoroughly revise the manuscript to correct all typo/grammatical/spelling errors.
The authors consider editing some sentences to be more appropriate, for example:
- Line 63: Besides hardware, wireless communication techniques in WSAN networks are also a significant issue
- Line 273: The second Task, the second, transmitters every time it is received from the nodes to gateway. Moreover, the last Task is optional and ensures the visual mechanisms on OLED for debugging.
- Line 284: Anyone can build apps without permission from big companies or the government
3 Authors should clearly define the terms LoRa and LoRaWAN used in manuscripts. Some places there is confusion between these two terms, for example line 94, line 95, line 156, line 160.
4 The author clarifies the impact of FreeRTOS on system performance. What is the size of this operating system? Is FreeRTOS really effective when applied to a small microcontroller like Atmega328P? Get some line updates for an explanation FreeRTOS applied in this system.
5 The author updates more information about the ESP-NOW protocol used in the system. State the reason for using this protocol in the proposed system, when LoRaWAN has a very optimal coverage distance.
6 Why do the authors suggest using a hybrid protocol between LoRa and LoRaWAN? Can the recommendation system fully utilize LoRaWAN? The author presents some discussion on this issue.
7 In the proposed figure, do outdoor nodes use an air quality improvement mechanism? Get some line updates for this issue.
8 The full-duplex relay node algorithm is ambiguous. Tasks have not been clearly shown their priority. The graphical representation of the sequential algorithm has not yet exploded the advantage of the parallel work of FreeRTOS.
9 Where is the Kalman filter used in this proposed system? Is it really necessary to use digital filters in the proposed model?

Reviewer 2 ·

Basic reporting

no comment and please see below

Experimental design

no comment and please see below

Validity of the findings

no comment and please see below

Additional comments

After reviewing this manuscript, the reviewer suggests the following corrections. After that, the final decision for this manuscript will be made:
* The authors check the abbreviations to ensure that they are consistent throughout in the manuscript. The authors correct some unclear sentences in the manuscripts, for example:
- Line 215 and 216: First, as mentioned, two LoRa modules’ simultaneous operation enables the system to expand without increasing communication delay.
- Line 219: … in half-duplex mode, the system performance will be improved
- Line 223: The downside of this 223 way of working is to increase communication delay. However, it can increase redundancy for the system
* What is the purpose of using the Blynk dashboard, while TagoIO already provides dashboards that can be displayed on Smartphones very well? The authors provide some reference to the Blynk dashboard.
* What communication class is the LoRa modules in the LoRaWAN system using? The authors present some information about this communication class.
* Add a few sentences in the conclusion to highlight the contributions of the work.
* Add some sentences describing the scalability of the clusters in the system. How many sensors is the system capable of supporting? How many nodes can the system scale up?
* In general, the modules in the whole manuscript are presented clearly, however, the author should add a complete model of the system to highlight the contribution of this paper.
* The tests in the EXPERIMENTATION AND RESULTS section are not clear. Discuss how these tests are performed and the time it takes to collect the data.
* Some system parameters have not been fully added to table 1, such as SF. Please check again and complete this table.

---

## Round 0.2 · Minor Revisions

The manuscript has improved significantly. However, the authors need to provide a table that shows a comparison with the state-of-the-art air quality monitoring techniques by using any key performance indicator.

Reviewer 2 ·

Basic reporting

The revision is now better than the previous version and good for reading.

Experimental design

The questions and objectives of this paper is now clear and well-defined. The experimental results can also validate what the authors find.

Validity of the findings

The contributions now are clear.

Additional comments

This paper is revised accordingly and can be accepted for publication. The reviewer appreciates the work.

Reviewer 3 ·

Basic reporting

The revised version of the paper is well written and organized and the authors perfectly reply to all the reviewer comments.

Experimental design

All good

Validity of the findings

No more work reqiured.

Additional comments

No comments

---

## Round 0.3 · accepted · Accept

The manuscript has improved significantly. I am satisfied with the improved revised version now.